# Voluntary Biosurveillance of *Streptococcus equi* Subsp. *equi* in Nasal Secretions of 9409 Equids with Upper Airway Infection in the USA

**DOI:** 10.3390/vetsci10020078

**Published:** 2023-01-20

**Authors:** Camilo Jaramillo-Morales, Kaitlyn James, Samantha Barnum, Wendy Vaala, Duane E. Chappell, Chrissie Schneider, Bryant Craig, Fairfield Bain, D. Craig Barnett, Earl Gaughan, Nicola Pusterla

**Affiliations:** 1William R. Pritchard Veterinary Medical Teaching Hospital, School of Veterinary Medicine, University of California, Davis, CA 95616, USA; 2Department of Obstetrics, Gynecology and Reproduction Biology, Massachusetts General Hospital, Boston, MA 02114, USA; 3Department of Medicine and Epidemiology, School of Veterinary Medicine, University of California, Davis, CA 95616, USA; 4Merck Animal Health, Madison, NJ 07940, USA

**Keywords:** *Streptococcus equi* subsp. *equi*, strangles, biosurveillance program, nasal secretions, qPCR detection, prevalence factors

## Abstract

**Simple Summary:**

Strangles caused by *Streptococcus equi* subsp. *equi* (*S. equi*) is considered one of the most prevalent and widely distributed infectious diseases in equids. Large epidemiological studies looking at prevalence factors associated with clinical disease are seldom reported. The present study aimed at describing selected prevalence factors of 9409 equids with acute onset of fever and respiratory signs tested for *S. equi* by qPCR and to determine the impact of vaccination on *S. equi* detection. A total of 715 horses (7.6%) tested qPCR-positive for *S. equi*, of which 226 horses had coinfections with common respiratory viruses (EIV, EHV-1, EHV-4, ERBV). Various prevalence factors were associated with *S. equi* qPCR-positive status, including season (winter and spring), use (competition and ranch/farm use), and clinical signs (nasal discharge, fever, lethargy, anorexia, and ocular discharge). Vaccination against *S. equi* was associated with a lower frequency of *S. equi* qPCR-positive index cases.

**Abstract:**

This study aimed to describe selected epidemiological aspects of horses with acute onset of fever and respiratory signs testing qPCR-positive for *S. equi* and to determine the effect of vaccination against *S. equi* on qPCR status. Horses with acute onset of fever and respiratory signs from all regions of the United States were included in a voluntary biosurveillance program from 2008 to 2020 and nasal secretions were tested via qPCR for *S. equi* and common respiratory viruses. A total of 715/9409 equids (7.6%) tested qPCR-positive for *S. equi*, with 226 horses showing coinfections with EIV, EHV-1, EHV-4, and ERBV. The median age for the *S. equi* qPCR-positive horses was 8 ± 4 years and there was significant difference when compared to the median age of the *S. equi* qPCR-negative horses (6 ± 2 years; *p* = 0.004). Quarter Horse, Warmblood, and Thoroughbred were the more frequent breed in this horse population, and these breeds were more likely to test qPCR-positive for *S. equi* compared to other breeds. There was not statistical difference for sex between *S. equi* qPCR-positive and qPCR-negative horses. Horses used for competition and ranch/farm use were more likely to test qPCR-positive for *S. equi* (*p* = 0.006). Horses that tested *S. equi* qPCR-positive were more likely to display nasal discharge, fever, lethargy, anorexia, and ocular discharge compared to horses that tested *S. equi* qPCR-negative (*p* = 0.001). Vaccination against *S. equi* was associated with a lower frequency of *S. equi* qPCR-positive status.

## 1. Introduction

Strangles caused by *Streptococcus equi* subsp. *equi* (*S. equi*) has been reported for centuries and it has been suggested that the spread of this highly contagious disease is a legacy from World War I [1,2]; nonetheless, large populations of horses around the world are still affected by this highly transmissible streptococcal disease, causing an economic burden for their owners. Strangles has been reported as the third-most-common upper respiratory tract infection in the USA after equine herpesvirus-4 (EHV-4) and equine influenza virus (EIV) [3]. The relevance of this disease is confirmed by contemporary research and the publication of two consensus statements by the American College of Veterinary Internal Medicine [4,5].

*Streptococcus equi*, a well-recognized and highly contagious microorganism with over 230 isolates sequenced worldwide, poses epidemiological challenges since various genotypes relate to outbreaks [6]. Vaccines are commercially available; however, due to the risk of vaccine-related immunological reactions, such as purpura hemorrhagica or abscess formation, horses with high serum titers to *S. equi* are not routinely vaccinated [7]. Association of strangles with other upper respiratory tract diseases has been poorly described in the literature, as well as association of vaccination status and development of natural disease.

Prevalence of strangles based on qPCR detection has been reported in the USA (6.4%) [3], the United Kingdom (4.6%, North Yorkshire) [8], specific regions in Brazil (2.3%, Rio Grande do Sul) [9], and Colombia (13.5%, Antioquia) [10]. Further, a prevalence of 1.6% based on culture has been reported from Western Canada [11]. Seroprevalence of *S. equi* by ELISA testing has been reported from various regions in South Africa [12], Israel [13], and Ethiopia [14] with 10.1%, 9.5%, and 8% respectively. There are scarce studies of demographic and prevalence factors associated with strangles. A previous study from the USA described prevalence factors in 49 horses with respiratory signs and qPCR detection of *S. equi* in nasal secretions [3]. Other studies have focused on clinical markers able to determine the risk of long-term silent carriers following infection [15].

Therefore, the aims of the present study were: (1) to describe the prevalence of *S. equi* in nasal secretions from 9409 horses with acute onset of fever and respiratory signs enrolled in a voluntary biosurveillance program for respiratory pathogens; (2) to describe demographic and clinical factors in *S. equi* qPCR-positive horses; and (3) to determine the impact of vaccination status on the detection of *S. equi*. Authors have hypothesized that the prevalence of *S. equi* is comparable to that previously reported, as would demographic characteristics, and that vaccinated horses would be less likely to test qPCR-positive for *S. equi*.

## 2. Materials and Methods

### 2.1. Study Population

The study population was composed of horses with acute onset of respiratory signs enrolled in a voluntary surveillance program from March 2008 to December 2020. Two hundred and sixty-one equine veterinary practices were enrolled in the biosurveillance program; the practices were located in various geographical regions (East n = 37, South n = 82, Midwest n = 45, West n = 97). Horses were sampled by equine veterinarians if the equid displayed at least one of the following clinical signs: acute onset of fever (T > 101.5 °F or > 38.5 °C) and/or respiratory signs, including nasal discharge (serous, mucoid, or purulent bilateral nasal discharge) and coughing. Information pertaining to additional clinical signs, such as lethargy, anorexia, ocular discharge, and limb edema, was collected as well. Sample collection was left at the discretion of each of the enrolled equine veterinarians with no case exclusion due to age, sex, or vaccination status.

### 2.2. Data Collection

The participant veterinarians were asked to fill out a questionnaire with information pertaining to signalment (age, breed, sex), intended use (competition, ranch/farm use, breeding, other use), history of transport, number of affected animals (single, multiple), vaccine history against *S. equi*, and clinical signs (fever, nasal discharge (serous, mucoid or purulent), cough, lethargy, partial or complete anorexia, ocular discharge (serous, mucoid, or purulent), and limb edema).

### 2.3. Sample Collection and Analysis

The attending veterinarians performed a physical examination and sampled the horses wearing disposable gloves; a nasal swab was collected using two 6” rayon-tipped swabs (Puritan Products Company LLC, Guilford, ME, USA). Samples were refrigerated and shipped overnight to the laboratory.

DNA purification from nasal swabs was performed on the day of sample arrival to the laboratory using an automated nucleic acid extraction system (QIAcube HT, Qiagen, Valencia, CA, USA) according to the manufacturer’s recommendations. Purified nucleic acids were assayed for the presence of *S. equi*, equine influenza virus (EIV), equine herpesvirus-1 (EHV-1), EHV-4, and equine rhinitis A and B viruses (ERVs) according to previously validated assays [3]. Further, in order to determine sample quality and efficiency of nucleic acid extraction, all samples were assessed for the presence of the housekeeping gene eGAPDH, as previously described [3].

### 2.4. Statistical Analysis

Demographic and clinical factors were compared between qPCR *S. equi*-negative and qPCR *S. equi*-positive status. Demographic factors included coinfections, breed, use, sex, age (analyzed continuously and categorized into 5-year increments), history of transport, number of affected horses on property, season of submission, and geographic region. Vaccination history (EHV-1/-4, EIV, and *S. equi*) and the presence of fever, nasal discharge, cough, lethargy, anorexia, ocular discharge, and limb edema were also compared between *S. equi* qPCR-positive and *S. equi* qPCR-negative horses. Parametric (chi-square and Student’s *t*-test) and non-parametric tests (Fisher’s exact and Mann–Whitney U test) were used as appropriate to compare categorical and continuous factors; a *p*-value < 0.05 was considered statistically significant. All statistical analyses were conducted in StataIC 16.0 (College Station, TX, USA).

## 3. Results

### 3.1. Prevalence of S. equi and Coinfections

Out of 9409 horses with acute onset of upper respiratory tract infection, 715 (7.6%) and 8694 horses tested qPCR-positive and qPCR-negative for *S. equi*, respectively. The most prevalent infection detected in nasal swabs from this horse population was EHV-4 (992 equids (10.5%)), followed by EIV (910 (9.7%)), ERBV (311 (3.3%)), EHV-1 (154 (1.6%)) and ERAV (12 (0.1%)). Coinfections were also detected in the reported horse population (Table 1). There was a statistically significant difference in horses testing qPCR-positive for EHV-4 (*p* = 0.03), EIV (*p* < 0.001), and ERBV (*p* < 0.001) and concurrent qPCR-positive status for *S. equi*. Horses that tested qPCR-positive against EHV-1 (*p* = 0.08) and ERAV (*p* = 0.30) were less likely to test qPCR-positive for *S. equi* (Table 1). Out of 6123 horses that had single infection, 408 (6.6%) tested qPCR-positive for *S. equi* in nasal swabs; on the other hand, out of 2409 horses that had multiple infections, 226 (9.4%) tested qPCR-positive for *S. equi*; there was a statistical difference between these two groups (*p* < 0.001). In 877 submissions, only one individual respiratory pathogen was tested (additional horses tested once the ethology of an outbreak was identified), out of which 81 (11.3%) horses tested qPCR-positive to *S. equi* (Table 1).

### 3.2. Demographics

The main breeds reported were Quarter Horse (44.4%), Warmblood (19.9%), Thoroughbred (13.5%), Arabian (8.6%), Paint Horse (5.7%), Pony/Miniature (4.6%), and draft breeds (3.2%); other breeds totaled 1895 horses. Quarter Horses were more likely to test qPCR-positive for *S. equi* compared to Warmbloods, Thoroughbreds, ponies, and other breeds (*p* < 0.001). Thoroughbreds were more likely to test qPCR-positive for *S. equi* compared to Paint horses, Arabians, draft horses, and ponies (*p* < 0.001); Warmbloods were more likely to test qPCR-positive for *S. equi* compared to draft horses and less likely to test qPCR-positive compared to Paint horses, Arabians, ponies, and other breeds (*p* < 0.001). Ponies were more likely than Arabians and draft horses and less likely than other breeds to test qPCR-positive for *S. equi*. Finally, Paint horses were less likely than ponies to test qPCR-positive for *S. equi* (Table 2).

The age of the study horses ranged from 3 months to 32 years. Median age for the *S. equi* qPCR-negative horses was 6 ± 2.1 years and for the *S. equi* qPCR-positive horses was 8 ± 4.1 years (*p* = 0.004); specific data associated with the age of the horses can be found in Table 2. Horses younger than 1 year of age were less likely to test qPCR-positive for *S. equi* compared to those between 1 and 4 years, 5 and 9 years, 10 and 14 years, and 15 and 19 years of age (*p* < 0.001). Horses between 1 and 4 years of age were less likely to test qPCR-positive for *S. equi* compared to horses between 5 and 9 years of age (*p* = 0.024). Horses between 5 and 9 years of age were more likely to test qPCR-positive for *S. equi* compared to horses aged 10–14 (*p* = 0.013) and > 20 years of age (*p* =0.002; Table 2).

Information on sex was available from 7902 horses with 3305 females and 4597 males (Table 2). Amongst females, 249 (7.5%) tested qPCR-positive for *S. equi* and 3056 tested qPCR-negative. Amongst males, 366 (7.9%) tested qPCR-positive for *S. equi* and 4231 tested qPCR-negative. There was no statistical difference for sex between *S. equi* qPCR-positive and qPCR-negative horses (*p* < 0.05).

Regarding the use of the horse, there were 3831(40.7%) competition horses, 3458 (36.7%) ranch/farm horses, 442 (4.7%) breeding horses, and 842 (8.9%) horses used for other purposes; the use was not reported for 836 (8.8%) horses (Table 2). Competition and ranch/farm horses were more likely to test qPCR-positive for *S. equi* (*p* = 0.006). From the 9409 horses with reported transportation history, 2456 (26%) were transported within 8 weeks. Amongst the horses with recent transportation history, 201 tested qPCR-positive for *S. equi* in nasal swabs and 2264 tested qPCR-negative. No statistically significant difference was detected (*p* = 0.14; Table 2).

### 3.3. Vaccination Status and Clinical Signs

Vaccination against *S. equi* was reported in 913 horses, from which 89 (9%) tested qPCR-positive for *S. equi* in nasal swabs (Table 3). Horses vaccinated against *S. equi* were less likely to test qPCR-positive for *S. equi* (*p* = 0.001). Immunization against EHV-1 and -4 and EIV was reported in 3054 and 3032 horses, respectively; there was no statistical difference between vaccination status against EHV-1/-4 and EIV and qPCR test results for *S. equi* (*p* > 0.05).

Horses that tested *S. equi* qPCR-positive in nasal swabs (715) were more likely to have the following clinical signs: nasal discharge (639 horses, 89%), fever (590, 83%), lethargy (539, 75%), anorexia (448, 63%), cough (368, 51%), and ocular discharge (205, 29%) compared to horses that tested *S. equi* qPCR-negative (*p* = 0.001; Table 3); limb edema occurred at the same frequency between horses testing *S. equi* qPCR-positive and qPCR-negative (*p* > 0.05). No significant differences (*p* > 0.05) between *S. equi* qPCR-positive with and without co-infection were observed for clinical signs.

### 3.4. Location and Seasonality

Horses in this study were grouped by geographic region. Amongst the 3370 horses with reported geographic region, 325 tested qPCR-positive for *S. equi*. There was no statistically significant difference between regions for horses testing positive or negative for *S. equi* by qPCR (*p* = 0.63; Figure 1). The location was unknown in 6039 horses; of those, 390 tested *S. equi* qPCR-positive.

Horses testing *S. equi* qPCR-positive were reported during all four seasons. Out of the 705 *S. equi* qPCR-positive horses, 197 (27.6%) tested positive in winter, 194 (27.1%) tested positive in spring, 124 (17.3%) tested positive in summer, and 119 (16.1%) tested positive in fall (Figure 2). There was a statistical difference between the horses that tested positive in summer/fall compared to those that tested positive in winter/spring (*p* < 0.001), having more *S. equi* qPCR-positive horses during winter/spring than during summer/fall. There was not a statistical difference between horses testing qPCR-positive in nasal swabs for *S. equi* during winter and spring (*p* = 0.66) or during summer and fall (*p* = 0.31).

## 4. Discussion

The present epidemiological study found a high prevalence of *S. equi* (7.6%) in nasal swabs from clinically ill horses in the USA; coinfections and association with EIV, ERBV, and EHV-4 were present in 31.6% of the *S. equi* qPCR-positive cases. This study also showed that vaccinated horses were less likely to test qPCR-positive for *S. equi*. As hypothesized, the prevalence found in this study was similar with the reported prevalence of *S. equi* from other countries, ranging from 1.66 to 13.5% by microbiological culture [11] or PCR [10]; nonetheless, the techniques for the sampling and detection of *S. equi* varied amongst the various studies [8,12,13,14]; thus, the results should be compared with caution.

In this study, a high rate of coinfection was detected between *S. equi* and EIV, and ERBV and EHV-4. Coinfections with various respiratory pathogens are well described and understood in cattle as bovine respiratory disease complex [16]. Viruses damage the respiratory epithelium, impairing both mucociliary clearance and phagocytic activity of macrophages and dendritic cells; hence, one could hypothesize that viral coinfections might predispose bacterial colonization of the respiratory tract. Because the severity of clinical disease between single infection and coinfections was not assessed, the impact of coinfections on disease expression for this horse population cannot be determined at the present time.

Quarter Horse, Warmblood, and Thoroughbred were the more frequent breeds in this horse population, and these breeds were more likely to test qPCR-positive for *S. equi* compared to other breeds. The present study also documented that competition horses were more likely to test qPCR-positive for *S. equi* than horses used for other purposes. Because the mentioned breeds are mainly used for competition, it is possible that such horses are at a higher risk of being in contact with clinically or subclinically infected *S. equi* horses.

Horses that had recent transportation history did not differ from those that did not travel when tested for *S. equi*. The results are in contrast with a study from Colombia, were transported horses within 3 months of onset of disease were more likely to have detectable *S. equi* in their guttural pouches [10]. Differences in sampling site (nasal secretions versus guttural pouch lavages) might explain the discrepant results.

Middle-aged horses were more likely to test qPCR-positive in nasal swabs for *S. equi* than other age groups. Middle-aged horses are more likely to be used for competition and are, therefore, more likely to become exposed to respiratory pathogens. The present age-related observation is in agreement with a retrospective study performed on the East coast of the USA with 108 horses, where the median age was 8 years [17], and with another study performed in Colombia where the median age of *S. equi*-positive horses was 6 years [10].

Nasal discharge, fever, lethargy, anorexia, cough, and ocular discharge were found associated with horses testing qPCR-positive for *S. equi* in the present study; those clinical signs are well known to be associated with strangles [18,19].

Vaccination against *S. equi* has been studied for a long time; however, due to the high genetic variation in these bacteria, standardization of the vaccines and their efficacy has been questioned. Killed and cell extract *S. equi* vaccines have shown moderate to excellent protection, when comparing vaccinated horses with those that had not been vaccinated [20,21]. Live-attenuated vaccines have been proven safe; however, the efficacy of those vaccines has been questioned [22,23]; moreover, live-attenuated vaccines do not allow for serological differentiation between vaccinated horses and horses with naturally occurring exposure [24]. In the present study, it was found that vaccination against *S. equi* significantly protected horses against testing *S. equi* qPCR-positive in nasal secretions. The data suggest a protective effect of *S. equi* vaccination in the study population. Unfortunately, information pertaining to the type of *S. equi* vaccine and administration route was unavailable, preventing us from drawing any comparative conclusion on the efficacy of the vaccine type.

Study limitations relate to the voluntary nature of the study and the lack of randomization of enrolled veterinary clinics. While the number of enrolled clinics by geographic region was in line with horse density, it may have impacted the demographic study population, therefore, affecting the various prevalence factors. It is, therefore, important to keep in mind that the data generated from this study are specific to the enrolled horse population. Further, the samples were solely obtained by nasal swabbing and not by guttural pouch lavage; the latter sample is known to be the most sensitive sampling technique [25]. Recent work has shown that a combination of sampling methods can increase the sensitivity and the ability to detect *S. equi* infection during different stages of infection [15]. Finally, because of the voluntary enrollment of the veterinary clinics, the index cases were not randomly distributed across the United States.

## 5. Conclusions

The prevalence of *S. equi* in nasal swabs of horses with respiratory signs from the USA remains high and suggests endemicity; middle-aged horses that compete are more likely to test positive for *S. equi* in nasal swabs when they present with acute onset of fever and respiratory signs. Finally, vaccination was associated with a lower frequency of *S. equi* qPCR-positive index cases.

## Figures and Tables

**Figure 1 vetsci-10-00078-f001:**
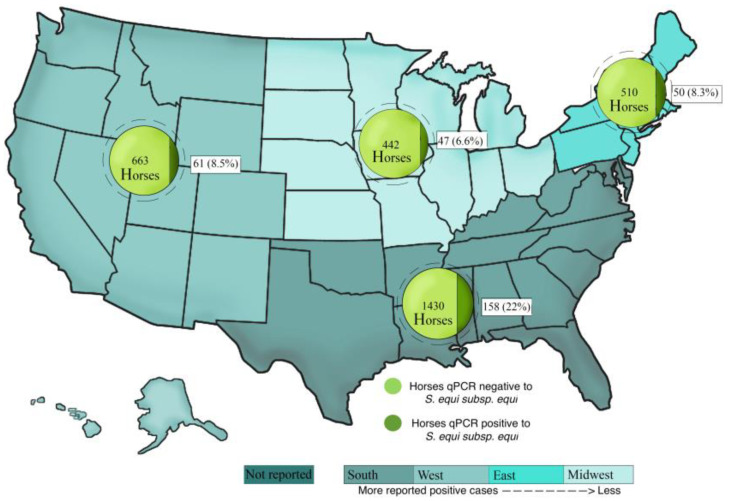
Four regions of the United States showing numbers and percentages of *S. equi* qPCR-positive and *S. equi* qPCR-negative horses. A total of 9409 equids with fever and respiratory signs were enrolled in the voluntary biosurveillance program.

**Figure 2 vetsci-10-00078-f002:**
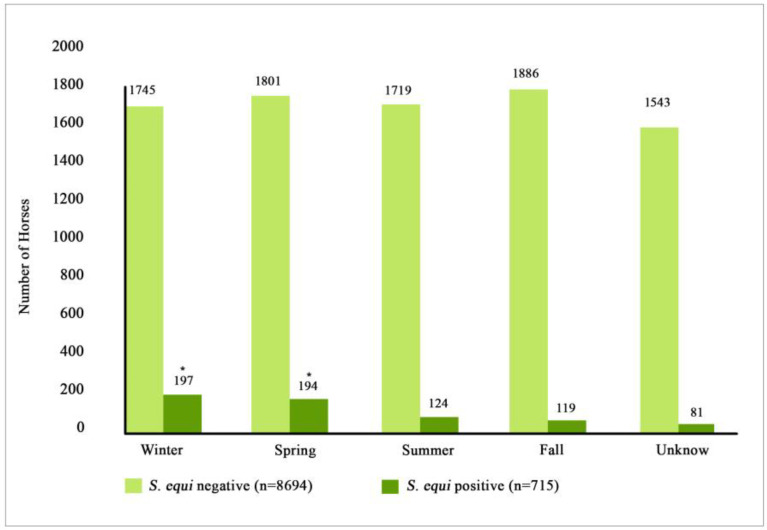
Seasonal distribution of *S. equi* qPCR-positive and *S. equi* qPCR-negative horses. Stars show significant statistical difference (*p* = 0.001). A total of 9409 equids with fever and respiratory signs were enrolled in the voluntary biosurveillance program.

**Table 1 vetsci-10-00078-t001:** Coinfections associated with *S. equi* qPCR-positive samples from 9409 equids with acute onset of fever and respiratory signs enrolled in a voluntary biosurveillance program.

PCR Results	*S. equi* qPCR-Negative Horses (8694)	*S. equi* qPCR-Positive Horses (715)	*p*-Value
Positive for EHV-4 (992)	934 (10.7%)	58 (8.1%)	0.037
Positive for EIV (910)	884 (10.2%)	26 (3.6%)	<0.001
Positive for ERBV (311)	261 (3.0%)	50 (7.0%)	<0.001
Positive for EHV-1 (154)	148 (1.7%)	6 (0.8%)	0.087
Positive for ERAV (12)	12 (0.1%)	0 (0.0%)	0.30
**Infection/coinfection**			<0.001
Single pathogen (6123)	5715 (65.7%)	408 (57.1%)	
Multiple pathogens (2409)	2183 (25.1%)	226 (31.6%)	
Not reported coinfection status (877)	796 (9.2%)	81 (11.3%)	

**Table 2 vetsci-10-00078-t002:** Demographic factors associated with *S. equi* qPCR-positive status in 9409 equids with fever and respiratory signs enrolled in a voluntary biosurveillance program.

Prevalence Factors	*S. equi* qPCR-Negative Horses (8694)	*S. equi* qPCR-Positive Horses (715)	*p*-Value
**Breed**			<0.001
Quarter Horse (3341)	3018 (34.7%)	323 (45.2%)	
Warmblood (1498)	1436 (16.5%)	62 (8.7%)	
Thoroughbred (1015)	987 (11.4%)	28 (3.9%)	
Arabian (645)	588 (6.8%)	57 (8.0%)	
Paint (427)	391 (4.5%)	36 (5.0%)	
Pony/Miniature (350)	291 (3.3%)	59 (8.3%)	
Draft Horse (238)	219 (2.5%)	19 (2.7%)	
Other Breed (1895)	1764 (20.3%)	131 (18.3%)	
**Age (years)**	6 ± 2.1	8 ± 4.1	
Less than 1 (1516)	1448 (16.7%)	68 (9.5%)	<0.001
1–5 (2345)	2159 (24.8%)	186 (26.0%)	
6–10 (2005)	1807 (20.8%)	198 (27.7%)	
11–15 (1413)	1308 (15.0%)	105 (14.7%)	
16–20 (859)	789 (9.1%)	70 (9.8%)	
Over 20 (573)	540 (6.2%)	33 (4.6%)	
No reported (698)	643 (7.4%)	55 (7.7%)	
**Sex**			0.24
Female (3305)	3056 (35.2%)	249 (34.8%)	
Male ^1^ (4597)	4231 (48.7%)	366 (51.2%)	
No reported (1507)	1407 (16.2%)	100 (14.0%)	
**Use**			0.006
Competition (3831)	3581 (41.2%)	250 (35.0%)	
Ranch/Farm horse (3458)	3157 (36.3%)	301 (42.1%)	
Breeding (442)	408 (4.7%)	34 (4.8%)	
Other Use (842)	779 (9.0%)	63 (8.8%)	
No reported (836)	769 (8.8%)	67 (9.4%)	
Transportation (2465)	2264 (26.0%)	201 (28.1%)	0.14

^1^ Male included geldings and stallions.

**Table 3 vetsci-10-00078-t003:** Clinical factors associated with *S. equi* qPCR-positive horses with acute onset of fever and respiratory signs enrolled in a voluntary biosurveillance program.

PCR Results	*S. equi* qPCR-Negative Horses (8694)	*S. equi* qPCR-Positive Horses (715)	*p*-Value
**Vaccine History**			
*S. equi* (913)	824 (9.5%)	89 (12.4%)	<0.001
EHV 1 and 4 (3054)	2839 (32.7%)	215 (30.1%)	0.11
EIV (3032)	2817 (32.4%)	215 (30.1%)	0.061
**Clinical Signs**			
Fever presence (6954)	6364 (73.2%)	590 (82.5%)	<0.001
Lethargy (6246)	5707 (65.6%)	539 (75.4%)	<0.001
Anorexia (5144)	4696 (54.0%)	448 (62.7%)	<0.001
Nasal discharge (6510)	5871 (67.5%)	639 (89.4%)	<0.001
Cough (4175)	3807 (43.9%)	368 (51.5%)	<0.001
Ocular discharge (2107)	1902 (21.9%)	205 (28.7%)	<0.001
Distal limb edema	806 (9.3%)	62 (8.7%)	0.87

## Data Availability

Data available on request due to privacy restrictions.

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
