# Peer review of "Voluntary Biosurveillance of *Streptococcus equi* Subsp. *equi* in Nasal Secretions of 9409 Equids with Upper Airway Infection in the USA"

_vetsci, 2023, doi:10.3390/vetsci10020078_

Round 1
Reviewer 1 Report
Dear Authors,
the paper is very interesting and add some new data to S. equi infection assessment. Some minor revision are needed.
Author Response
Unfortunately, the reviewer did not add the specific minor suggestions or comments to his review.
Reviewer 2 Report
REVIEWER COMMENTS FOR MANUSCRIPT Vetsci-2145199
The authors are reporting a descriptive biosurveillance study to describe prevalence of S. equi in nasal secretions of horses with acute onset fever and/or respiratory signs, factors related to qPCR-positivity, and the impact of S. equi vaccination status on test-positivity.
MAJOR ISSUES
1. Manuscript reports a descriptive study and did not control for potential confounding. This should be addressed in the discussion section as a limitation of this study.
2. Figures and tables should clearly indicate the study population – these are horses that presented to veterinary hospitals with acute onset fever and/or respiratory symptoms – this goes to the generalizability of study results.
3. Manuscript reports that this is a non-random sample. Need to specify the impact this may have on results and interpretations.
SPECIFIC COMMENTS
1. Line 88 – Need to clarify what is meant by an ‘index case.’ Does this mean that only 1 horse was sampled from each farm, owner, outbreak?
2. Line 90 – Define nasal discharge.
3. Line 96 – How was the participant veterinarian questionnaire administered? Was data checked for accuracy?
4. Line 100 – Define anorexia, ocular discharge, nasal discharge.
5. Line 115 – 124 – Did authors control for multiple comparisons in data analyses?
6. Table 2 – All factors report an omnibus p-value, but then ‘age’ reports an additional p-value. Recommend being consistent in reporting, and, if keeping this additional p-value, indicate what the additional p-value is for.
7. Line 276 – 278 – How does the limitation of a non-random sample impact these results and interpretations?
Author Response
Reviewer 2.
The authors are reporting a descriptive biosurveillance study to describe prevalence of S. equi in nasal secretions of horses with acute onset fever and/or respiratory signs, factors related to qPCR-positivity, and the impact of S. equi vaccination status on test-positivity.
MAJOR ISSUES
- Manuscript reports a descriptive study and did not control for potential confounding. This should be addressed in the discussion section as a limitation of this study.
The authors agree with the reviewer that the study did not enroll cases at random. First of all, selected veterinary clinics were enlisted into the biosurveillance program and second, the enrollment of index cases was voluntary. While the selection of veterinary clinics by region reflect horse density (i.e. more clinics were enrolled in areas with high horse density), the choice of submitting samples and questionnaires by the veterinarian was purely voluntary. The authors have added this issue into the limitations of the study. The bias may have affected the population demographic, hence affecting the prevalence factors. It is important to emphasize that the data generated from this study is specific to the population of horses tested, this latter statement relates to almost any field study.
- Figures and tables should clearly indicate the study population – these are horses that presented to veterinary hospitals with acute onset fever and/or respiratory symptoms – this goes to the generalizability of study results.
The authors thank the reviewer for bringing up this issue and have added information pertaining to the study population in the legends for Tables and Figures.
- Manuscript reports that this is a non-random sample. Need to specify the impact this may have on results and interpretations.
The lack of randomization in veterinary clinics may have biased the study population. While the study was voluntary, the veterinary clinics took advantage of this program and hopefully did enroll all cases presented to them with acute onset of fever and respiratory signs. While each clinic and each geographic region has its own specific horse population, the lack of randomization in clinic enrollment may have impacted the study population, therefore affecting the various prevalence factors. The authors have emphasized that the data generated from this study is specific to the population of horses tested.
SPECIFIC COMMENTS
- Line 88 – Need to clarify what is meant by an ‘index case.’ Does this mean that only 1 horse was sampled from each farm, owner, outbreak?
Index case means sick horse presenting to a veterinarian. To prevent any confusion, the term index case was replaced by horse.
- Line 90 – Define nasal discharge.
Nasal discharge included bilateral serous, mucoid or purulent discharge.
- Line 96 – How was the participant veterinarian questionnaire administered? Was data checked for accuracy?
The questionnaire was filled out by the attending veterinarians at the time the patient was examined and the biological samples collected. While the information pertaining to each of the questionnaires was not cross-referenced with clinical records, clinics from patients with missing information on the submitted questionnaire (lack of listed rectal temperature and other missing information) were contacted by phone in order to determine if the information was not entered or unknown.
- Line 100 – Define anorexia, ocular discharge, nasal discharge.
The clinical terms have been further characterized as requested by the reviewer.
- Line 115 – 124 – Did authors control for multiple comparisons in data analyses?
Comparisons were performed for each of the selected prevalence factors between two groups (S. equi qPCR-positive and S. equi qPCR-negative horses).
- Table 2 – All factors report an omnibus p-value, but then ‘age’ reports an additional p-value. Recommend being consistent in reporting, and, if keeping this additional p-value, indicate what the additional p-value is for.
The first value of p=0.004 reflects a difference in age between the S. equi qPCR-positive and qPCR-negative population. The second value of p<0.001 shows that horses less than one year of age were less likely to test qPCR-positive for S. equi compared to those between 1-4 years, 5-9 years, 10-14 years, and 15-19 years of age. The table has been adjusted to prevent any confusion.
- Line 276 – 278 – How does the limitation of a non-random sample impact these results and interpretations?
The clinics enrolled in the biosurveillance program were not randomized, meaning that equine veterinary practices were selected from the client directory of Merck Animal Health. This may have biased clinics towards equine clinics only, versus mixed animal clinics or small animal clinics with a small emphasis on large animals. It would be difficult to randomized cases as all horses presenting with fever and respiratory signs would benefit from diagnostic procedures. It would have been interesting to enroll non-diseased horses originating from the same premises as clinically diseased horses (control population) in order to determine the rate of subclinical disease.
Reviewer 3 Report
vetsci-2145199-peer-review
Voluntary biosurveillance of Streptococcus equi subsp. equi in
nasal secretions of 9,409 equids with upper airway infection in the USA
lines 22 and 32- ...”coinfections with common respiratory viroses”... What viroses? EHV-4 131 (992 equids (10.5%)), followed by EIV (910 (9.7%)), ERBV (311 (3.3%)), EHV-1 (154 (1.6%)) 132 and ERAV (12 (0.1%))
Lines 24 and 40 - Is it possible to identify the type and method of vaccination? Because the immune response may be different according these ones.
Line 65 – instead “Soul”, write “Sul”
General comments: The statistical analysis, Parametric (chi-square and Student’s t-test) and non-parametric tests (Fisher’s exact and Mann Whitney U test) were used to compare demographic and clinical factors between qPCR S. equi-negative and qPCR S. equi-positive status. I ask if in this statistical analysis is possible to include the clinical signs among S. equi-negative/positive qPCR status and viral coinfection.
Author Response
Voluntary biosurveillance of Streptococcus equi subsp. equi in nasal secretions of 9,409 equids with upper airway infection in the USA
lines 22 and 32- ...”coinfections with common respiratory viroses”... What viroses? EHV-4 131 (992 equids (10.5%)), followed by EIV (910 (9.7%)), ERBV (311 (3.3%)), EHV-1 (154 (1.6%)) 132 and ERAV (12 (0.1%))
The missing information on the co-infection with respiratory viruses has been added in the abstract.
Lines 24 and 40 - Is it possible to identify the type and method of vaccination? Because the immune response may be different according these ones.
Unfortunately, the reported information did not include type and administration route for the S. equi vaccines. In the USA, there are two types of S. equi vaccines available, i.e. a killed-adjuvanted bacterin vaccine given intramuscularly and a modified-live vaccine given intranasally. Information pertaining to this limitation has been added in the discussion.
Line 65 – instead “Soul”, write “Sul”
The wording has been changed as suggested by the reviewer.
General comments: The statistical analysis, Parametric (chi-square and Student’s t-test) and non-parametric tests (Fisher’s exact and Mann Whitney U test) were used to compare demographic and clinical factors between qPCR S. equi-negative and qPCR S. equi-positive status. I ask if in this statistical analysis is possible to include the clinical signs among S. equi-negative/positive qPCR status and viral coinfection.
No significant differences (P > 0.05) between S. equi qPCR-positive with and without co-infection were observed for clinical signs. The missing information has been added in the manuscript.